# *Francisella tularensis* enters a double membraned compartment following cell-cell transfer

**Shaun P Steele\*, Zach Chamberlain, Jason Park, Thomas H Kawula**

School of Global Animal Health, Washington State University, Pullman, United States

**Abstract** Previously, we found that phagocytic cells ingest bacteria directly from the cytosol of infected cells without killing the initially infected cell (Steele et al., 2016). Here, we explored the events immediately following bacterial transfer. *Francisella tularensis* bacteria acquired from infected cells were found within double-membrane vesicles partially composed from the donor cell plasma membrane. As with phagosomal escape, the *F. tularensis* Type VI Secretion System (T6SS) was required for vacuole escape. We constructed a T6SS inducible strain and established conditions where this strain is trapped in vacuoles of cells infected through bacterial transfer. Using this strain we identified bacterial transfer events in the lungs of infected mice, demonstrating that this process occurs in infected animals. These data and electron microscopy analysis of the transfer event revealed that macrophages acquire cytoplasm and membrane components of other cells through a process that is distinct from, but related to phagocytosis.
DOI: https://doi.org/10.7554/eLife.45252.001

## Introduction

Antigen presenting cells (APCs) acquire and present immunogenic material to mount both innate and adaptive immune responses to pathogens. The current paradigm focuses on APCs acquiring immunostimulatory material from phagocytosis of extracellular microbes or through the recognition of microbial molecules by surface receptors on the plasma membrane. However, there is evidence that this model is incomplete. It assumes that APCs only acquire intracellular pathogens during intermittent times when the microbes are extracellular. But immune cells can also acquire infectious material and antigen directly from infected cells through a few different pathways.

Here, we focus on a process where APCs directly acquire infectious material and generic cytosolic proteins from the cytosol of infected cells (*Steele et al., 2016*; *Perez et al., 2017*; *Cambier et al., 2017*; *Utter et al., 2017*; *Ramirez and Sigal, 2002*). This occurs with a wide range of microbes and several lines of evidence suggest that cell-cell transfer of bacteria plays a critical role in pathogenesis *in vivo* [1–3]. But the underlying mechanism of bacterial transfer is unclear. Without understanding the mechanism, it is difficult to manipulate this bacterial transfer process to directly assess its role during infections.

We previously found that macrophages acquire microbes and cytosolic content from neighbouring cells and the acquired bacteria exploit this process to sustain infection (*Steele et al., 2016*). Several possibilities for how the bacteria move between cells have been proposed. Here, we sought to identify how cell-cell transfer occurs. We found that macrophages phagocytose a small portion of a neighbouring cell. Importantly, only a piece of the initially infected cell is engulfed and the donor cell survives after a small portion is phagocytosed (*Steele et al., 2016*). To differentiate this process from phagocytosis of the entire infected cell, we propose the term merocytophagy (Greek for partial cell eating). After merocytophagy, the bacteria and cytosolic material acquired from the donor cell

**\*For correspondence:**
shaun.steele@wsu.edu

**Competing interests:** The authors declare that no competing interests exist.

enter a unique, double-membraned endosome that is composed in part by the donor plasma membrane.

## Results

### Macrophages phagocytose portions of infected cells to acquire both bacteria and cytosolic material

We previously found that bacterial transfer from infected to uninfected BMDMs requires cell-cell contact (*Steele et al., 2016*). To resolve how bacteria transfer between cells, we performed transmission electron microscopy (TEM) of bone marrow derived macrophages (BMDMs) during bacterial transfer. For this assay, we mixed BMDMs that had been infected for 20 hr with *Francisella tularensis* and uninfected BMDMs. The donor and recipient cell were identified based on the cell-cell interaction in the image.

By TEM, the recipient cell appeared to engulf a small protrusion of the donor cell (*Figure 1A and B*). Notably, the donor cell fragment was contiguous with the cytosol of the host in the initial slices

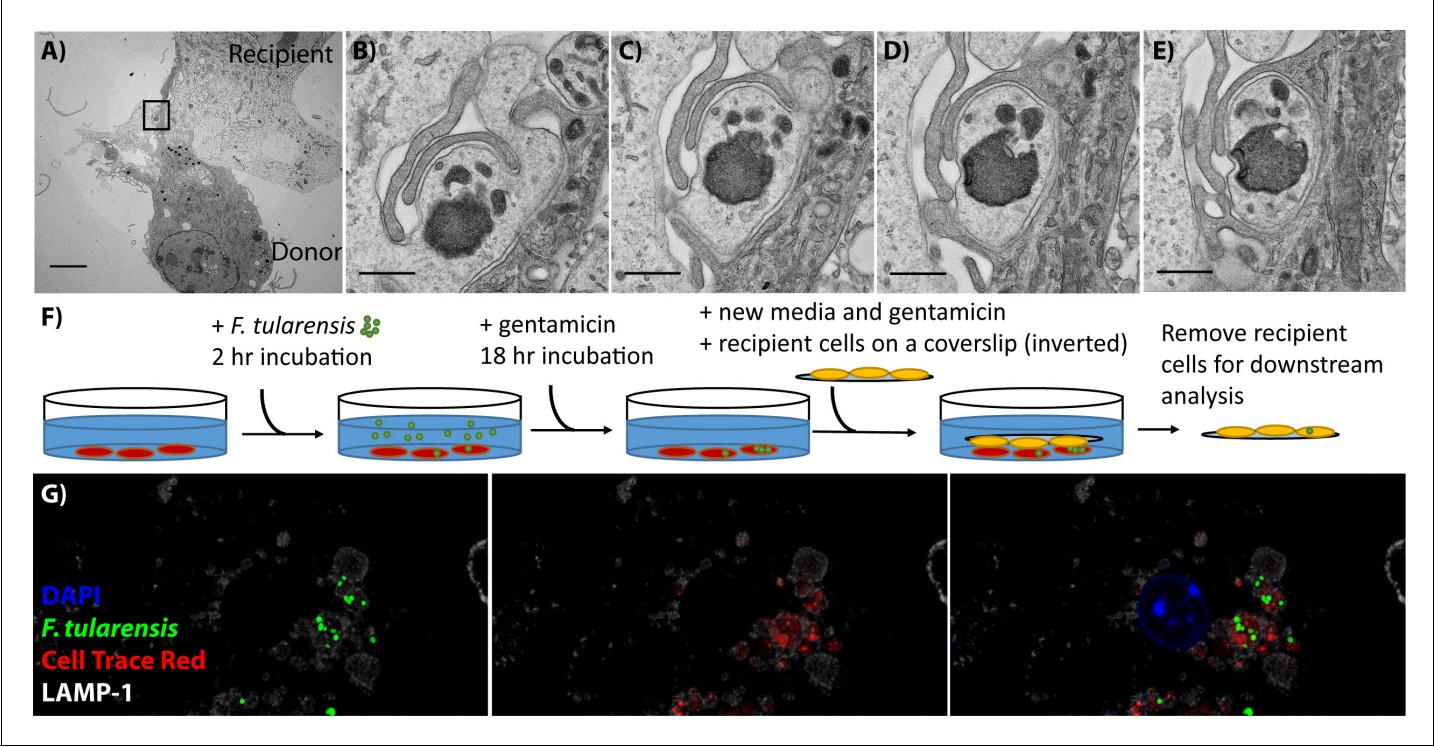

**Figure 1.** BMDMs acquire bacteria and cytosolic content from neighbouring cell via phagocytosis. (**A**) Transmission electron microscopy of a donor and recipient BMDM. The less electron dense cell is the donor cell in this instance. The scale bar represents 5 µm. (**B–E**) Higher magnification images of the black box in panel A. Each panel is a sequential slice through the same region. The scale bar represents 500 nm. (**F**) A diagram of the synchronized transfer assay. Recipient cells are seeded onto a coverslip, inverted onto the infected cells and then the coverslip is removed to purify the recipient cells. (**G**) Representative confocal microscopy image of a recipient cell after bacterial transfer. This image indicates that bacteria and cytosolic content are both acquired together. The different images represent different combinations of stains and the complete overlay. *F. tularensis* (green), transferred cytosolic protein (Cell Trace Red) (red), LAMP-1 (white) and DAPI (blue). An example donor cells is depicted in *Figure 1—figure supplement 1*.

DOI: https://doi.org/10.7554/eLife.45252.002

The following figure supplement is available for figure 1:

**Figure supplement 1.** Representative image of a donor cell in cytosolic transfer assay.

DOI: https://doi.org/10.7554/eLife.45252.003

but was surrounded by protrusions from the recipient in sequential slices. These data indicate that BMDMs phagocytose small portions of their neighbours.

The material that the macrophage acquired appears to include a *F. tularensis* bacterium based on shape and electron density. *F. tularensis* is typically identified in TEM images by of the characteristic electron translucent capsule surrounding the bacteria, which this bacterium lacks (*Steele et al., 2013*) (Example in Figure 5). The fragmentation of the bacterium and lack of capsule suggests that this particular bacterium may be getting degraded during the transfer process or a killed bacterium is being transferred between cells. Cell-cell transfer is a host-mediated process. So killed bacteria, and potentially even bacterial fragments, are fully capable of transferring between macrophages. It is important to note that in the case of *F. tularensis*, the majority of bacteria are viable following cell-cell transfer (*Steele et al., 2016*).

## Host proteins transfer with the bacteria

The TEM results suggest that cytosolic material transferred with the bacteria (*Figure 1B*). To test this observation, we labelled infected donor cells with Cell Trace Red, a dye that labels intracellular proteins by binding to free amines (*Figure 1—figure supplement 1*). We then inverted a coverslip seeded with unstained, uninfected BMDMs onto the dyed, infected cell population. We let the cells incubate like this for 30 min so that cell-cell transfer could occur between the two populations. We then removed the coverslip, which is almost exclusively 'recipient' cells (*Figure 1F*) (0.13 ± 0.23% of cells are infected donors that migrated to the coverslip, three independent experiments, 500 cells per experiment analyzed, mean ± SD).

We stained recipient cells for LAMP-1 and assessed if Cell Trace Red and bacteria were within the same vacuole following bacterial transfer. We found that most *Francisella* containing vacuoles (FCVs) also contained Cell Trace Red labelled protein from the donor cell cytosol *(Figure 1G)*. From these results, we conclude that both host cytosolic proteins and bacteria are acquired within the same vacuole following bacterial transfer.

## *F. tularensis* enters and escapes an endocytic compartment following cell-cell transfer

Our results indicate that BMDMs phagocytose portions of live cells but does not reveal what happens to the acquired material following transfer. Phagocytosis of extracellular *F. tularensis* leads to co-localization of bacteria with the early endosomal marker EEA-1. The *F. tularensis* containing phagosome matures, which results in co-localization with the late endosomal marker LAMP-1 (*Craven et al., 2008*). The bacteria then rupture and escape the phagosome, entering the cytosol where they replicate. We were interested in whether FCVs follow a similar maturation process after cell-cell transfer.

Using the assay described in *Figure 1F* with modified co-incubation times, we found that *F. tularensis* bacteria were typically located in EEA-1$^+$ vacuoles at early time points post-transfer (*Figure 2A and C*). These FCVs matured into LAMP-1$^+$ vacuoles over time (*Figure 2B and D*). Interestingly, the kinetics of LAMP-1 maturation and escape are virtually identical between cell-cell transfer and phagocytosis of extracellular bacteria (*Figure 2D*). There was a slight delay in EEA-1 maturation following bacterial transfer compared to extracellular bacteria (*Figure 2C*), but this apparent delay was likely due to much higher variability in the timing of infections through cell-cell transfer, rather than delayed maturation. These data suggest that *Francisella* interactions with the host are similar regardless of entry route.

## The Francisella type VI secretion system is required for post-transfer endosomal escape, but not for cytosolic replication

Following uptake of extracellular bacteria, *F. tularensis* requires a type VI secretion system (T6SS) to escape the phagosome (*Clemens et al., 2018*). Due to the similarities in the vacuole maturation kinetics between extracellular uptake and bacterial transfer, we hypothesized that the T6SS was also required for escape from the recipient cell endosomes after bacterial transfer. To test this hypothesis, we needed a strain that was fully functional in the initially infected cells, but was unable to escape the endosome in the recipient BMDM. To accomplish this goal, we put one of the T6SS

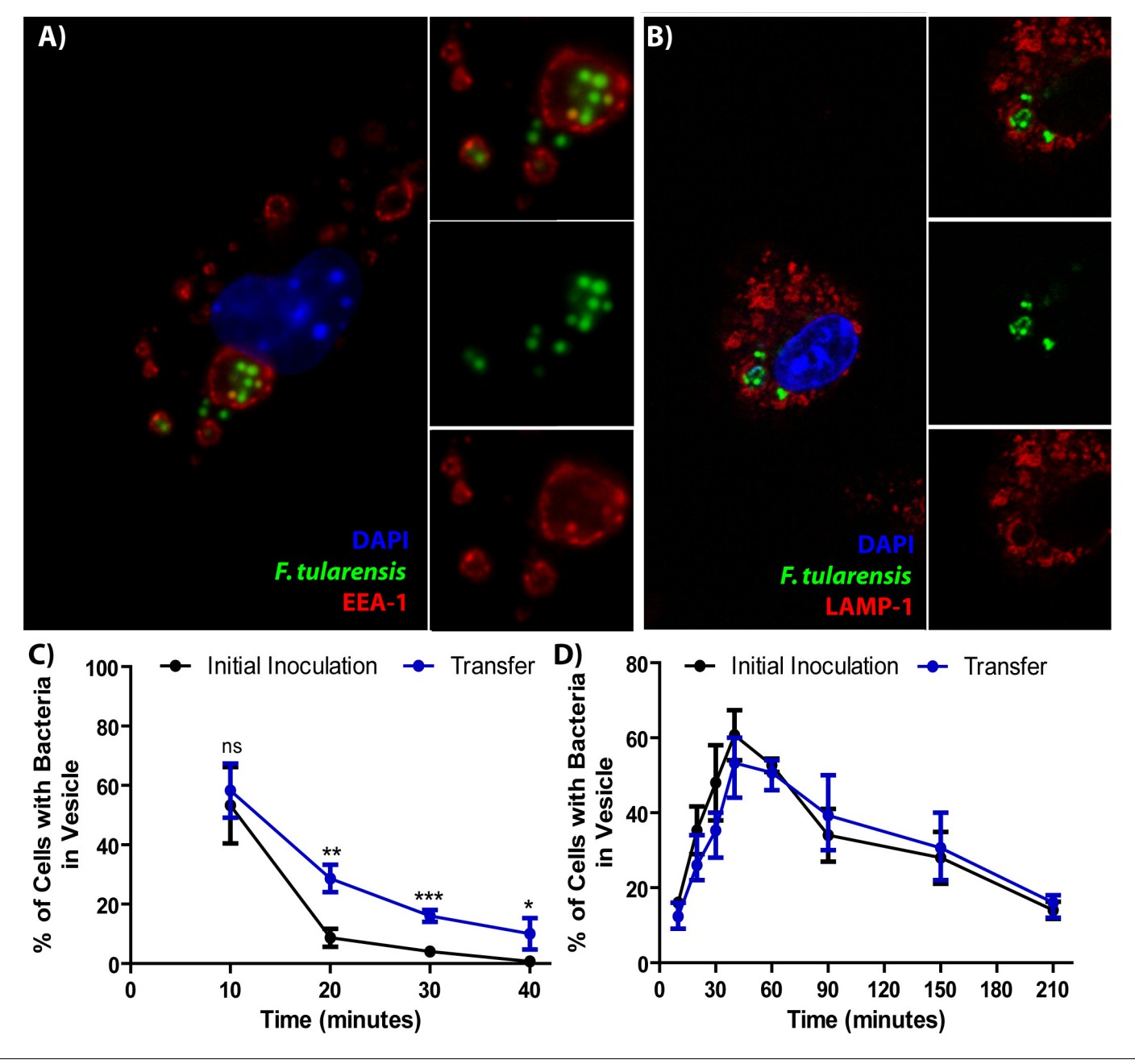

**Figure 2.** *F. tularensis* enters the endocytic pathway in recipient cells after cell-cell transfer. (A) Representative image of *F. tularensis* (green) inside an EEA-1 (red) positive vacuole 10 min after synchronized cell-cell transfer. (B) Representative image of *F. tularensis* (green) inside a LAMP-1 (red) positive vacuole 1 hr after synchronized cell-cell transfer. (C–D) The percentage of cells with at least one bacterium enclosed inside of (C) EEA-1 or (D) LAMP-1 positive vacuoles. The black line represents a conventional infection where the BMDMs phagocytose extracellular bacteria. The blue line represents purified recipient BMDMs after bacterial transfer. From three independent experiments with 50 infected cells counted per experiment per time point. Student t-test. Mean ± standard deviation. *p<0.05, **p<0.01, ***p<0.001. ns- no statistical significance. No time points were significantly different for LAMP-1 co-localization.

DOI: https://doi.org/10.7554/eLife.45252.004

structural protein genes (FTL_0119) under anhydrous tetracycline (ATc) induction in a FTL_0119 deletion background. In this strain, the T6SS is only able to form in the presence of ATc.

We infected BMDMs with the T6SS inducible strain after growing the bacteria overnight in broth containing ATc. The inoculation and subsequent infection were in media lacking ATc. Under these

conditions, the T6SS strain escaped the initial phagosome and replicated in the cytosol at nearly the same rate as wild-type (*Figure 3A*, *Figure 3—figure supplement 1*). These data indicate that the T6SS is largely dispensable for *F. tularensis* intracellular replication so long as the mutant escapes the initial phagosome. This is crucial because different amounts of replication likely impact the amount of cell-cell transfer.

Normally, *Francisella* escapes the FCV over time (*Figure 3B*). But following cell-cell transfer with the T6SS inducible strain, the FCV remained intact for at least 6 hr (*Figure 3B*). The phagosomal escape defect is similar between the post-transfer FCV in the T6SS inducible strain and impaired escape in the initial phagosome following inoculation with the uninduced T6SS or a marker-less, in-frame *dotU* deletion strain. These data indicate that *Francisella* uses the same machinery to escape both vacuoles regardless of the route of entry.

In addition to showing similarities between phagosomes and post-transfer FCVs, the T6SS inducible strain enables us to accumulate FCVs in recipient BMDMs over time (*Figure 3B*). This makes it more likely that we can detect bacteria in FCVs post transfer. Additionally, the bacteria serve as a marker to identify vacuoles post-transfer. We will use this strain in later experiments to test FCV formation during cell-cell transfer by TEM and during mouse infections.

## Cell-cell transfer enhances bacterial transmission compared to extracellular uptake

*F. tularensis* is thought to primarily infect macrophages when the cells phagocytose extracellular bacteria. Our previous results suggest that *F. tularensis* exploits cell-cell transfer to infect cells and that this process may be common in vivo (*Steele et al., 2016*). It is unclear if macrophages preferentially acquire bacteria through these infection routes, so we tested if BMDMs acquired more bacteria if the bacteria were extracellular compared to acquisition of bacteria from the cytosol of infected cells.

For these experiments, we normalized the amount of bacteria in each sample. Donor BMDMs had approximately 1 million intracellular bacteria, so the control group was infected with 1 million free living bacteria using a synchronized infection where the bacteria were centrifuged onto the BMDMs. To synchronize transfer, we used the same assay as described in *Figure 1F*. During a brief inoculation, BMDMs acquired 10-fold more bacteria by phagocytosing bacteria from inside of neighbouring, infected BMDMs than uptake of extracellular bacteria (*Figure 4A*). Approximately 5% of the total bacteria transferred from the infected to uninfected BMDMs. These results suggest that

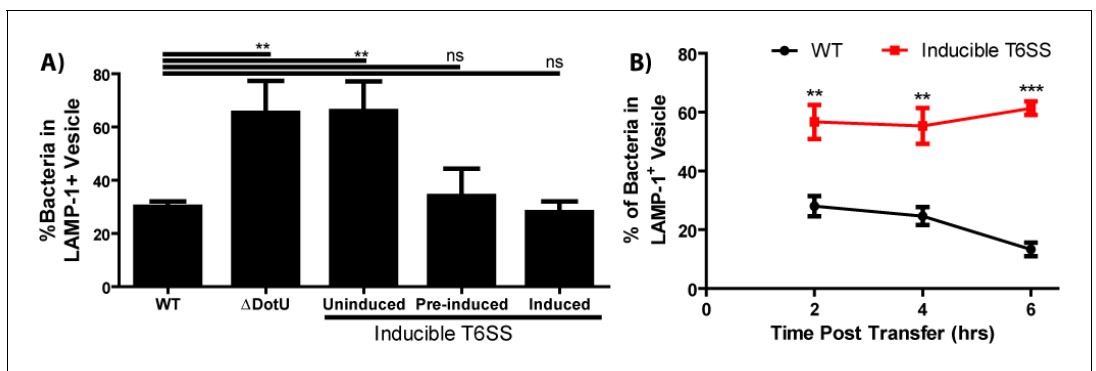

**Figure 3.** The type VI secretion system is required for Francisella escape from the phagosome following cell-cell transfer. (**A**) The percent of bacteria enclosed in a LAMP-1 positive vacuoles 3 hr after a synchronized infection with extracellular bacteria. These data represent the ability of the inducible type VI secretion system (T6SS) strain to escape the initial phagosome during a conventional infection under the inducible expression condition. (**B**) The percent of bacteria enclosed in LAMP-1 positive vacuoles after cell-cell transfer. Under conditions where the inducible strain is not producing the T6SS, the bacteria remain largely trapped while the wildtype strain continues to escape the phagosome over time. From three independent experiments with 50 infected cells counted per experiment per time point. Mean ± standard deviation. Panel A used a One-way Anova with Dunnett post-test. Panel B used a Student t-test to compare between time points. **p<0.01, ***p<0.001, ns – not significant.

DOI: https://doi.org/10.7554/eLife.45252.005

The following figure supplement is available for figure 3:

**Figure supplement 1.** *The* F. tularensis type VI secretion system is dispensable for intracellular growth following phagosomal escape.
DOI: https://doi.org/10.7554/eLife.45252.006

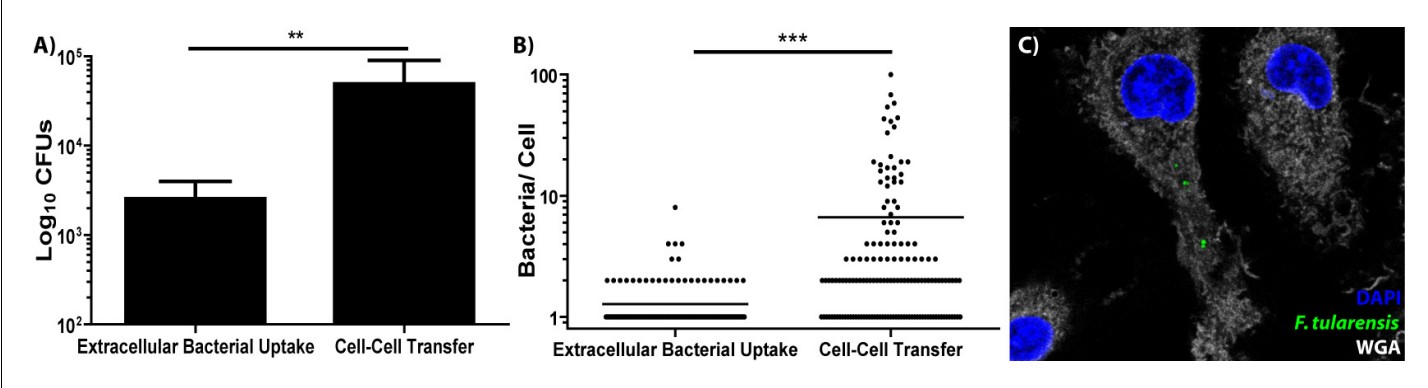

**Figure 4.** BMDMs acquire significantly more bacteria via bacterial transfer than phagocytosis of extracellular bacteria. (A) Colony forming units of intracellular bacteria 2 hr after synchronized infection or purified recipient cells after synchronized cell-cell transfer. three independent experiments performed in triplicate. Mean ± standard deviation. (B) The number of bacteria per cell in either BMDMs infected with extracellular bacteria or recipient BMDMs infected via bacterial transfer 2 hr post inoculation. three independent experiments with 50 infected cells counted per sample per experiment. Each data point represents an infected cell. Bar represents the mean. (C) Representative image of a recipient BMDM. The bacteria are depicted in green, DAPI in blue and the plasma membrane (wheat germ agglutinin) in white. Student t-test. **p<0.01, ***p<0.0001.
DOI: https://doi.org/10.7554/eLife.45252.007

bacteria migrate between cells much better through cell-cell transfer than through phagocytosis of extracellular *F. tularensis* bacteria.

A small population of infected cells migrates to the recipient population (0.13 ± 0.23%), which will slightly skew our colony forming unit results (three independent experiments, 500 cells examined per experiment, mean ± SD). To account for this, we validated these results by microscopy. We labelled the recipient BMDM plasma membranes with biotin before the two populations were mixed and used the synchronized transfer assay (*Figure 1F*). After isolating the recipient cells, we stained them with fluorescent streptavidin to ensure they were not contaminating donor cells. Only fully biotin labelled recipient cells were included in the microscopy analysis. We found that recipient BMDMs acquired significantly more bacteria per cell than BMDMs exposed to extracellular *F. tularensis* (*Figure 4B and C*). Taken together, cell-cell transfer of bacteria increased the number of bacteria that spread because recipient cells typically acquired multiple bacteria at the same time.

## *F. tularensis* enters a distinctive vacuole following cell-cell transfer

Part of the reason cell-cell transfer was so much more efficient at transferring bacteria is that several bacteria entered the same FCV (*Figure 5A*). Following phagocytosis of extracellular bacteria, the vast majority of cells have a single bacteria per vacuole (*Figure 5—figure supplement 1A*). In contrast, almost half of recipient cells with bacteria in a LAMP-1 positive vacuole have multiple bacteria within the same vacuole, with about 20% having more than five bacteria in the same vacuole (*Figure 5—figure supplement 1A*). Recipient cells often take several bites of the same donor cell, so the same cell often has multiple vacuoles containing bacteria (for example, *Figure 2A/B* and *Figure 4C*). Thus, recipient BMDMs acquired several *F. tularensis* bacteria simultaneously.

More bacteria per vacuole likely means more bacterial effectors were present to modify the vacuole, possibly increasing bacterial escape kinetics. If more bacteria within the same FCV increased phagosomal escape, the percentage of cells with multi-bacterial FCVs should decrease over time. Instead, we found that the percent of LAMP-1$^+$ vesicles containing multiple bacteria was consistent across the time points that we examined (*Figure 5—figure supplement 1B*). These data indicate that having multiple bacteria in a vacuole does not substantially alter escape kinetics. It also suggests that vacuoles with several bacteria are not intrinsically different from vacuoles with a single bacterium.

For these assays, we used the live vaccine strain of *F. tularensis*. However, we observed identical structures following cell-cell transfer of the highly virulent Schu S4 strain (*Figure 5—figure supplement 2*). Thus, these results are broadly applicable to virulent *F. tularensis*.

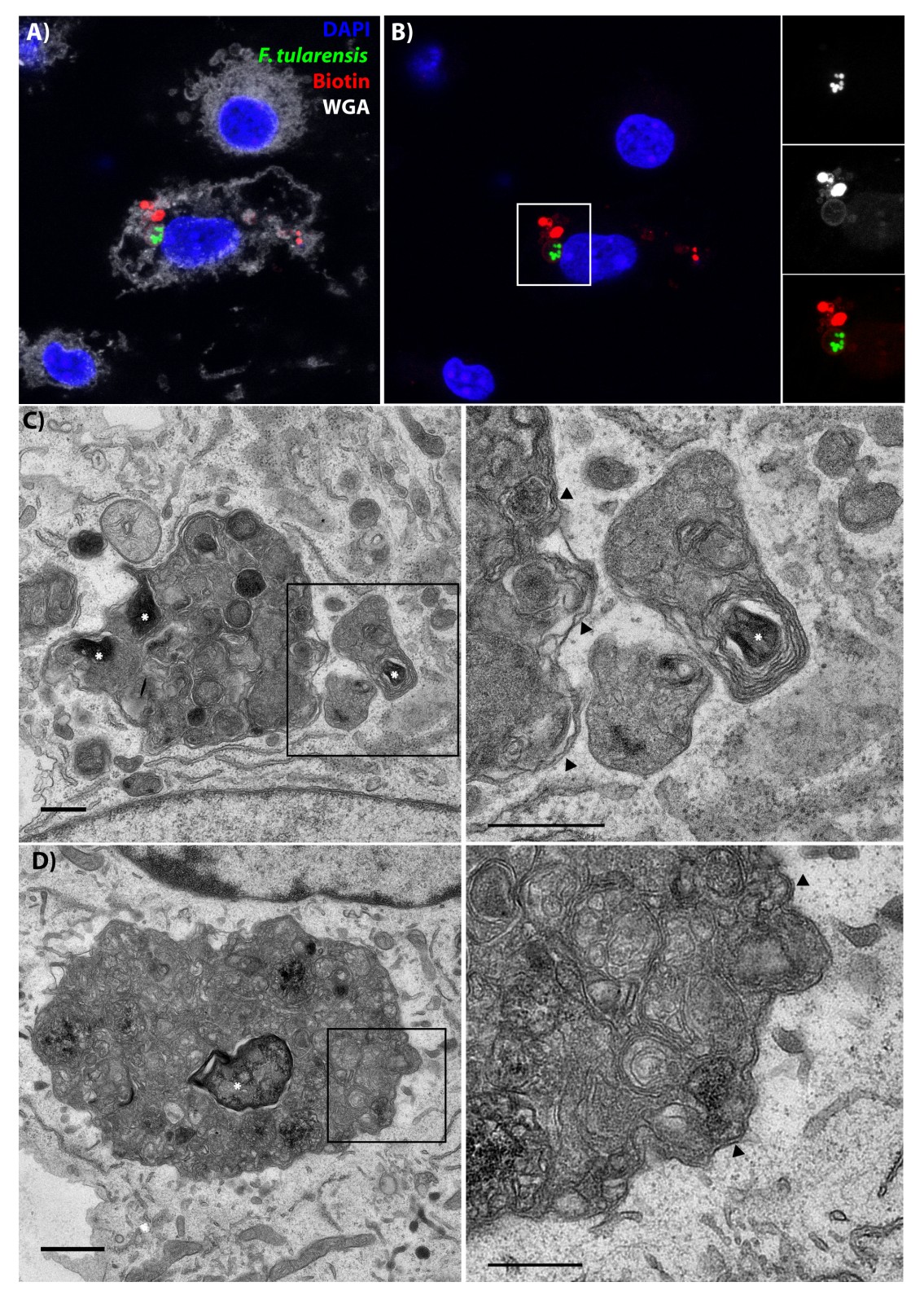

**Figure 5.** Following cell-cell transfer, Francisella is enclosed within a multi-membraned vacuole with one membrane originating from the donor plasma membrane. (**A**) Representative image of a recipient cell following bacterial transfer. The plasma membrane of the initially infected donor cell was labelled with biotin and the purified recipient cells were permeabilized and stained with fluorescent streptavidin (red). (**B**) The same image as panel A without the wheat germ agglutinin staining for the plasma membrane. The images on the right are higher magnifications of the white box. An example

*Figure 5 continued on next page*

*Figure 5 continued*

donor cells is depicted in *Figure 5—figure supplement 3* (C–D). Two representative transmission electron micrograph images of inducible T6SS bacteria inside of double membrane structures in purified recipient cells. The images on the right are higher magnifications of the boxed region for each respective image. Arrows denote double membranes, white stars denote bacteria. The scale bars are all 500 nm. Sequential slices and related structures in the same cell for Panel C are available in *Figure 5—figure supplement 6*.

DOI: https://doi.org/10.7554/eLife.45252.008

The following figure supplements are available for figure 5:

**Figure supplement 1.** Vacuoles containing several bacteria form following cell-cell transfer.
DOI: https://doi.org/10.7554/eLife.45252.009
**Figure supplement 2.** *Formation of FCVs in* the *F. tularensis* Schu S4 strain.
DOI: https://doi.org/10.7554/eLife.45252.010
**Figure supplement 3.** Representative images of donor cells in membrane transfer assays.
DOI: https://doi.org/10.7554/eLife.45252.011
**Figure supplement 4.** Transmission electron micrograph of a recipient BMDM that acquired wildtype bacteria.
DOI: https://doi.org/10.7554/eLife.45252.013
**Figure supplement 5.** FCVs containing several bacteria require cell-cell contact.
DOI: https://doi.org/10.7554/eLife.45252.014
**Figure supplement 6.** Transmission electron micrograph of a recipient BMDM that acquired several bacteria.
DOI: https://doi.org/10.7554/eLife.45252.012

## Transferred bacteria are enclosed in a multi-membranous structure

In *Figure 1B*, the donor cell plasma membrane is enclosed inside of the plasma membrane of the recipient BMDM. This result suggests that vacuoles formed through phagocytosis of intact cells should have two membranes. We hypothesized that the transferred material would be inside of a structure that resembles an autophagosome, but is made from membranes of two different cells. The plasma membrane of the recipient cell should enclose the structure, just like a typical phagosome. Inside of that should be a portion of the donor cell plasma membrane that was pinched off during transfer. The transferred cytosolic content would be at the core of this structure.

We first tested if bacteria were enclosed in vacuoles containing plasma membrane from the donor cell. To do this, we biotinylated the proteins on the plasma membrane of donor, infected cells. We then synchronized bacterial transfer to uninfected BMDMs and purified the recipient BMDMs. We permeabilized the cell and stained with streptavidin, which will bind to biotinylated proteins that were on the infected donor cell plasma membrane prior to cell-cell transfer. This procedure allows us to track what happens to the donor cell plasma membrane following cell-cell transfer. We found that structures with multiple *F. tularensis* bacteria were within a vacuole that is at least partially derived from the plasma membrane of the originally infected donor cell (*Figure 5A and B*, donor cell example in *Figure 5—figure supplement 3*).

If phagocytosis of the donor cell occurs, then there should be at least two membranes surrounding the transferred bacteria. To test this, we purified recipient cells using the synchronized transfer assay described *Figure 1F* and examined them by TEM. Since wildtype bacteria will escape the FCV, we primarily used the T6SS inducible strain for our TEM studies. *F. tularensis* bacteria are the dark, electron dense structures with a small electron translucent clearing around them.

The inducible T6SS strain was only found inside of multi-membraned structures in all of the infected recipients that we identified (*Figure 5C and D*). Similar to our confocal analysis, multiple bacteria were within the same vacuole and recipient BMDMs often contained several distinct vacuoles with bacteria (*Figure 5—figure supplement 4*). Notably, we were able to find wildtype bacteria in similar, multi-membranous structures at 40 min post-transfer (*Figure 5—figure supplement 4*). As expected, only a subset of wildtype bacteria were found inside of vacuoles due to phagosomal escape. From these results, we conclude that the FCV is a double-membraned structure composed of membrane from two different cells.

The morphology of these FCVs is unusual. The T6SS inducible bacteria were found within complex, membranous structures that most closely resemble residual bodies (*Novikoff and Shin, 1978*). Residual bodies are the undigested remnants of phagolysosomes or autophagolysosomes (*De Duve and Wattiaux, 1966*). Since the mutant cannot escape, the vacuole likely matures down the normal degradative pathway. In comparison, TEM of wildtype *F. tularensis* indicate that the

bacteria are in a more electron dense, less membranous compartment. This is likely due to modifications of the vacuole by the T6SS, which is essential for *F. tularensis* to escape the vacuole.

## Cell-Cell contact is required for FCVs containing multiple bacteria to form

A possible alternative explanation for the formation of FCVs containing multiple bacteria is phagocytosis of extracellular vesicles that contain several bacteria. In theory, this structure would also form a double membraned compartment after phagocytosis that could enclose several bacteria. To assess the likelihood that extracellular vesicles explain our phenotype rather than contact dependent cell-cell transfer, we analyzed how many cells had FCVs containing several bacteria when the infected and uninfected populations were physically separated.

We only observed FCVs containing several bacteria under conditions where cell-cell contact was possible (assay from *Figure 1F*). When infected and uninfected BMDMs were physically separated by 2–3 millimetres using a Transwell membrane, very few recipient BMDMs became infected and we did not find a single recipient cell with multiple bacteria in the same LAMP-1$^+$ vacuole in any of the experiments (*Figure 5—figure supplement 5*). In this setup, bacteria and extracellular vesicles can pass through the membrane, but whole cells from the different populations could not touch. In contrast, uninfected cells that were able to physically touch infected cells had readily observable multi-bacterial FCVs. Thus, cell-cell transfer needs to occur for recipient cells to acquire multiple bacteria within the same FCV.

## FCVs containing multiple bacteria occur in a mouse infection model

To test if multi-bacterial FCVs form *in vivo*, we intranasally inoculated mice with the inducible T6SS strain. The lungs were harvested 24 hr post inoculation and a single cell suspension of cells was allowed to adhere to a coverslip for 1 hr. The cells were then washed, fixed with paraformaldehyde and stained for LAMP-1. We then examined the adherent cells *ex vivo* by microscopy.

We readily found multi-bacterial clusters in LAMP-1$^+$ vacuoles in lung cells *ex vivo*, similar to our results using cultured BMDMs *(Figure 6)*. Based on the similarities with our *in vitro* results, cell-cell transfer of bacteria occurs during *F. tularensis* infections in the lung and is not a cell culture phenomenon.

It is difficult to determine how bacteria infect cells *in vivo* because there are several different potential routes that likely all occur. To estimate how much cell-cell transfer may be occurring *in vivo* during *F. tularensis* infections, we quantified the number of cells with T6SS mutant bacteria clustered within the same LAMP-1$^+$ vacuole. We excluded cells where bacterial growth occurred because these were most likely the initially infected cells (example in *Figure 6—figure supplement 1*). We found that 19.02 ± 2.93% of the recipient cells had three or more bacteria within the same LAMP-1 positive vacuole (mean ± SD, from 4 experiments with two mice pooled per sample, 45 putative recipient cells identified).

Importantly, this estimate does not include any transfer events where only one or two bacteria transferred at a time. Additionally, only about 70% of the T6SS mutant bacteria were found in LAMP-1 positive vacuoles *in vitro*, even though the mutant was trapped within the FCV (*Figure 3B*). Based on these conservative criteria, a minimum of about 1 in five infected recipient cells become infected via cell-cell transfer in the lung during *F. tularensis* infection.

## Discussion

*F. tularensis* and several other intracellular pathogens transfer directly between cells (*Steele et al., 2016*; *Perez et al., 2017*; *Cambier et al., 2017*; *Utter et al., 2017*). Here, we found that macrophages phagocytose portions of a living cells upon cell-cell contact and the acquired material goes through typical phagosomal maturation. The acquired material is found in double membrane vacuoles that often contain several bacteria, which makes the resulting FCV distinct from phagosomes formed from phagocytosis of extracellular material (*Figure 7*).

Several pathogens rely on secretion systems throughout their intracellular life cycle and are unable to properly regulate the host if the function is disrupted (*Smith et al., 2016*; *Klein et al., 2017*). As a control for our studies, we assessed the impact of the T6SS in *F. tularensis* following phagosomal escape. We found that that the T6SS had a negligible effect on intracellular growth as

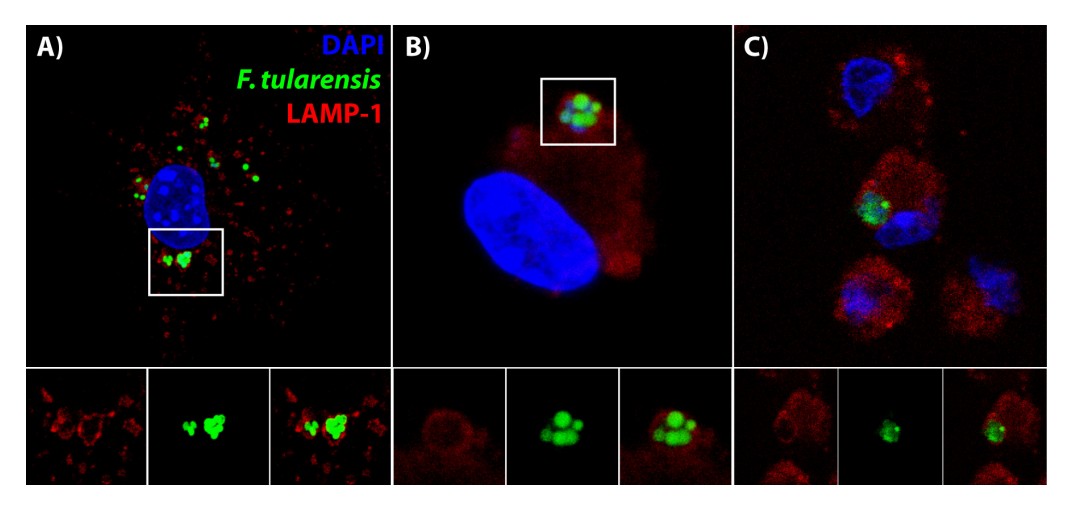

**Figure 6.** Bacteria are in distinctive vacuoles following cell-cell transfer in both BMDMs and lung cells *ex vivo*. (A) Representative image of a LAMP-1 stained recipient BMDM following bacterial transfer. (B–C) Representative images of LAMP-1 stained lung cell *ex vivo*. Note the similarity in structures between the known recipient BMDM in Panel A and the adherent infected cells *ex vivo*. *F. tularensis* is depicted in green, LAMP-1 in red and DAPI in blue.

DOI: https://doi.org/10.7554/eLife.45252.015

The following figure supplement is available for figure 6:

**Figure supplement 1.** Example of donor cell *in vivo*.

DOI: https://doi.org/10.7554/eLife.45252.016

long as the bacteria escape the initial phagosome. This is consistent with previous reports that the T6SS is dispensable for intracellular replication of *F. tularensis* (*Steele et al., 2014*; *Meyer et al., 2015*). In contrast, the attenuated *Francisella novicida* strain appears to require the T6SS for intracellular replication (*Wu et al., 2015*). It is unclear why the role of the T6SS varies between the different strains, but could be related to *F. novicida* containing two genes, *pdpD* and *anmK*, in its pathogenicity island that are functionally deleted in the live vaccine strain (LVS).

*F. tularensis* has previously been found within double membrane vacuoles of infected cells that resemble what we observed (*Checroun et al., 2006*). The assumption at the time was that the bacteria had re-entered autophagic vacuoles following replication in the cytosol. But more recent reports indicate that *F. tularensis* is very rarely targeted by autophagy (*Steele et al., 2013*; *Chong et al.,*

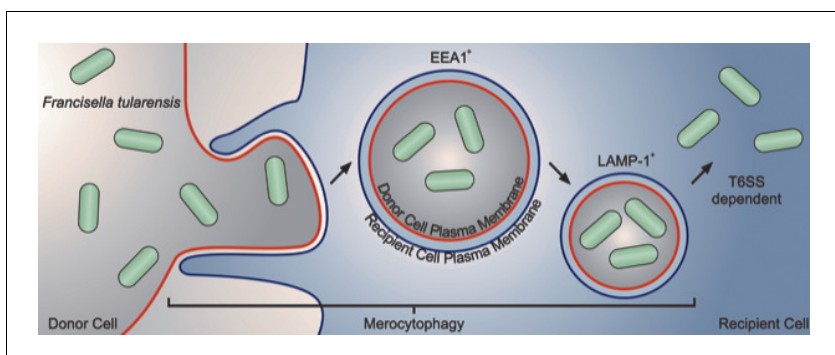

**Figure 7.** Graphical Summary. *F. tularensis* transfers between cells via phagocytosis of a small portion of a cell upon cell-cell contact, which we have termed merocytophagy. Following merocytophagy, the acquired cytosolic material enters a double membraned compartment. Each layer of membrane is derived from a different cell. When *F. tularensis* migrates between cells via merocytophagy, it escapes the vacuole using its type VI secretion system.
DOI: https://doi.org/10.7554/eLife.45252.017

*2012*; *Case et al., 2014*). Instead, our data suggests that at least a subset of these structures may have resulted from cell-cell bacterial transfer.

Similar structures have also been observed with other intracellular pathogens. When *Listeria monocytogenese* is propelled by actin into neighbouring cells, the bacteria are enclosed in a double membrane structure through a process termed paracytophagy (*Robbins et al., 1999*). One of the membranes is derived from the donor plasma membrane, and one is from the recipient. A key feature of paracytophagy is that it is a bacteria driven phenomenon. *F. tularensis* does not co-localize with actin or have homologs to known actin manipulating proteins (*Steele et al., 2016*). Instead, bacterial transfer of *F. tularensis* is host mediated and phagocyte specific, unlike paracytophagy which results from the activity of bacterial effector proteins (*Steele et al., 2016*; *Robbins et al., 1999*). Thus, the underlying transfer mechanism is different between the two processes, but the resulting structure is similar.

In our previous report, we found that bacterial transfer strongly correlated with a phenomenon termed trogocytosis (*Steele et al., 2016*). Trogocytosis is the transfer of plasma membrane proteins between cells in a way that the proteins remain functional in the recipient cell. Recent work suggests that trogocytosis strongly correlates with bacterial transfer but is not causative (data not shown). Part of the impetus for these experiments was to identify distinctive features that are specific to cell-cell transfer of bacteria. Trogocytosis was observed in some of our assays, but not explicitly tested or quantified. Based on our results, bacteria within multi-bacterial, double membrane endosomes directly indicate that bacterial transfer occurred.

Immune mediated phagocytosis of intracellular pathogens appears to be a widespread phenomenon (*Steele et al., 2016*; *Perez et al., 2017*; *Cambier et al., 2017*; *Utter et al., 2017*). This study suggests that bacterial transfer occurs when immune cells phagocytose a portion of intact cells. We propose the term merocytophagy to describe host-mediated phagocytosis of small portions of live cells without killing the donor cell. The resulting vacuole from merocytophagy is made from at least two membranes with one of the membranes originating from the donor cell plasma membrane. The resulting structure appears to be a degradative vacuole due to LAMP-1 association and that the vacuoles mature into a residual body when the bacteria are unable to escape. *F. tularensis* enters the merocytophagy pathway following bacterial transfer, but is able to escape with bacterial effectors secreted through the T6SS.

Our proposed term of merocytophagy designates that the acquired cellular material is in a degradative pathway. This distinguishes it from trogocytosis, which appears to be a plasma membrane-centric event. Some models of trogocytosis suggest the material is ingested, similar to merocytophagy, but the donor cell plasma membrane is then returned to and incorporated into the recipient cell plasma membrane (*Dopfer et al., 2011*). In this scenario, trogocytosis likely leads to the exocytosis of at least some of the material acquired from the cytosol of other cells. We speculate that the strong correlation between bacterial transfer and trogocytosis that we previously observed is because these processes initially acquire material from neighbouring cells using the same machinery. After acquisition, the material may be sorted and sent into different pathways, with trogocytosis being analogous to endocytic recycling and merocytophagy functioning similarly to endocytic degradation.

We focused on a bacterium that escapes the phagosome in this study, but pathogens that modify the vacuole may have a very different host-pathogen response. The double membrane in particular may pose problems for bacteria that modify and reside in vacuoles. Our studies point to this process being host mediated, so there must be a benefit to the immune response. Future studies are needed to resolve how the host benefits from this process.

## Materials and methods

**Key resources table**

| Reagent type (species) or resource | Designation | Source or reference | Identifiers | Additional information |
|---|---|---|---|---|

*Continued on next page*

*Continued*

| Reagent type (species) or resource | Designation | Source or reference | Identifiers | Additional information |
|---|---|---|---|---|
| Antibody | EEA-1 (goat monoclonal) | Santa Cruz Biosciences | clone N19, catalog number sc-6415 | 1:50 dilution, permeabilized in 0.1% saponin, 2% FBS in humidifier at 37 degrees Celcius for 30 minutes. Cells cannot be fixed for longer than 7 minutes prior to staining. |
| Antibody | LAMP-1 (rat monoclonal) | Developmental Studies Hybridoma Bank | Clone 1D4B | 1:500 dilution after stock mixed with 50% glycerol, permeabilized in 0.1% saponin, 2% FBS in humidifier at 37 degrees Celcius for 30 minutes |
| Antibody | Anti-rat secondary (goat monoclonal) | ThermoFisher Scientific | catalog number 26-4826-82 | 1:500 dilution in 0.1% saponin, 2% FBS in humidifier at 37 degrees Celcius |
| Antibody | Anti-goat secondary (donkey monoclonal) | ThermoFisher Scientific | catalog number PA1-28662 | 1:500 dilution in 0.1% saponin, 2% FBS in humidifier at 37 degrees Celcius |
| Strain, strain background (Mus musculus, female) | 6–10 week old female C57Bl/6J | Jackson Labs | catalog number 000664 | |
| Strain, strain background (*Francisella tularensis,* Live Vaccine Strain) | *F. tularensis* or LVS | CDC | | |
| Strain, strain background (*Francisella tularensis,* Schu S4) | *F. tularensis* Schu S4 | BEI Resources | | |
| Strain, strain background (*Francisella tularensis,* inducibleT6SS mutant) | T6SS mutant | this paper | | *F. tularensis* strain with an in-frame, markerless deletion of FTL_0119 (*dotU*) that is complemented with a pEDL17 plasmid containing *dotU* under the tetracycline inducible promoter and GFP on a constitutive promoter |

*Continued on next page*

*Continued*

| Reagent type (species) or resource | Designation | Source or reference | Identifiers | Additional information |
|---|---|---|---|---|
| Strain, strain background (*Francisella tularensis,* Empty Vector Control) | WT | this paper | | wildtype LVS that is complemented with a pEDL17 plasmid containing a kanamycin resistance cassette under the tetracycline inducible promoter and GFP on a constitutive promoter |
| Strain, strain background (*Francisella tularensis,* Empty Vector in Δ*dotU*) | Δ*dotU* | this paper | | A FTL_0119 deletion in LVS that is complemented with a pEDL17 plasmid containing a kanamycin resistance cassette under the tetracycline inducible promoter and GFP on a constitutive promoter |
| Chemical compound, drug | anhydrous tetracycline | Cayman Chemicals | catalog number 100009542 | |

## Antibodies and critical reagents

The EEA-1 antibody (N-19, sc-6415) and LAMP-1 antibody (1D4B) were acquired from Santa Cruz Biotechnologies (Dallas, Texas) and the Developmental Studies Hybridoma Bank (University of Iowa, Iowa City, Iowa) (*Hughes and August, 1981*) respectively. The secondary antibodies used were anti-Rat (cat# 26-4826-82) and anti-goat (cat# PA1-28662) from ThermoFisher Scientific (Waltham, Massachusetts).

## Cell culture

Bone Marrow derived Macrophages were differentiated and used as previously described (*Steele et al., 2016*). BMDMs were seeded the night before infection at 200,000 cells per well in a 12 well plate. For all microscopy studies, the imaged BMDMs were seeded onto an acid treated coverslip.

## Bacterial growth

*F. tularensis* live vaccine strain and Schu S4 were grown on chocolate agar supplemented with isovitalex for 3 days. The day before infection, the cells were seeded into Chamberlain's defined media. The culture was shaken at 37°C overnight, typically 16–18 hr.

## Synchronized infection by extracellular bacteria

All bacterial infections where uptake of extracellular bacteria was tested were performed by synchronizing the infection. Briefly, cells were chilled on ice and had cold media containing bacteria at either a multiplicity of infection of 5 (experiments comparing bacterial uptake to cell-cell transfer) or 100 (all other assays). The cells were placed into a chilled centrifuge and spun for 5 min at 300x g. The cells were then heat shocked in a 37° water bath and incubated for 30 min. Following incubation, the media was exchanged with media containing 10 ug/ml gentamicin.

## Synchronized transfer assays

BMDMs were seeded directly into a 12 well plate or placed on coverslips. 20 hr post inoculation, the media in both wells was replenished with fresh media containing 10 ug/ml gentamicin and the uninfected BMDMs on the coverslip were inverted onto the infected BMDMs. After a 30 min incubation, the coverslip was removed and placed back in the original well. The cells were then harvested at the

indicated time. This procedure had a 0.13 ± 0.23% frequency of infected donor cells attaching to the coverslip (three independent experiments, 500 cells examined per experiment, mean ± SD).

## Inducible type VI secretion system

The tetracycline inducible promoter containing plasmid pEDL17 was modified to constitutively express GFP and express either kanamycin resistance (empty vector controls) or DTL_0019 (*dotU*) (*LoVullo et al., 2012*). Genes under the control of the tetracycline inducible promoter were induced by 250 ng/ml of anhydrous tetracycline (ATc) (Cayman Chemicals, Ann Arbor, Michigan) in Chamberlains Defined Media for 18 hr prior to inoculating cells. In all induction experiments, ATc was added to the overnight growth media for all samples except the uninduced control. ATc was removed prior to inoculation of host cells unless otherwise indicated. *pEDL17-GFP empty vector had a slight but consistent delay in phagosomal escape compared to the pkk214-GFP plasmid used in most of our studies.*

## Quantification of bacteria after Cell-Cell transfer

BMDMs were synchronously infected with either infected BMDMs in the presence of gentamicin or extracellular bacteria. Extracellular bacteria were added at a multiplicity of infection of 5. This MOI was chosen because the donor BMDMs averaged five bacteria per cell based on bacterial colony forming units at 20 hr post inoculation and the initial number of BMDMs seeded.

After a 30 min incubation, the slide with infected cells in the cell-cell transfer sample was returned to its original well and 10 ug/ml of gentamicin in fresh media was added to all samples. At 2 hr after the start of transfer, the cells were scraped from the plate, lysed by vortexing, serially diluted and quantified.

## Transfer of donor plasma membrane

BMDMs were infected for 19 hr with GFP-expressing *F. tularensis*. 10 ug/ml gentamicin added 2 hr post inoculation. The infected cells were labelled with biotin using an EZ-link sulfo-NHS- biotin labelling kit following the manufacturer's protocol (ThermoFisher Scientific). The cells were washed with complete media and incubated for 30 min after biotin labelling. A coverslip with uninfected, unlabelled cells were placed onto the infected cells for 1 hr. The cells were then lifted off, fixed with 4% paraformaldehyde, washed with PBS followed by 50 mM ammonium chloride, stained with wheat germ agglutinin (cat# W32466 ThermoFisher Scientific) and imaged by confocal microscopy.

## Microscopy

Microscopy results were acquired using a Leica DM4000 upright fluorescence microscope. For a bacterium or bacterial cluster to be counted as within an EEA-1 or LAMP-1 positive vacuole, the fluorescent marker had to completely surround the bacteria, as in *Figure 2A and B*. Bacteria that colocalized with these markers but where the markers did not form a coherent vacuole were not counted as being within a vacuole.

Occasionally, BMDMs will have bacteria in multiple vacuoles or some in vacuoles and some not within a vacuole. For these cells, if any of the bacteria were contained in a LAMP-1 or EEA-1 positive vacuole, they were counted as having bacteria inside of a vacuole. Likewise, for determining the percentage of cells with multiple bacteria within individual LAMP-1 positive vacuole, any cell with multiple bacteria in the same vacuole was counted, even if bacteria were not in a vacuole elsewhere in the cell.

Representative confocal images were acquired using a Leica SP-8 microscope and the same samples were used for representative images and data analysis where applicable.

## Mouse cell to cell transfer analysis

Mice were anesthetized with avertin and intranasally inoculated with 10,000 colony forming units of the inducible T6SS strain. The bacteria were grown in chamberlains defined media containing 250 ng/ml ATc overnight and 250 ng/ml of ATc was added to the PBS containing bacteria that the mice were inoculated with.

At 24 hr post inoculation, the lungs were harvested. Single cell suspensions were made by finely chopping the lung with scissors and incubating the lung pieces in five units per ml of dispase (Stem

Cell Technologies, Cambridge Massachusetts) for 30 min at room temperature. The lung homogenate was put through a 40 μm cell strainer to isolate individual cells. The sample was centrifuged at 250x g for 5 min and the supernatant was removed. RPMI with 10% FBS and supplemented with non-essential amino acids, Glutamax, and sodium pyruvate was added to the cells for the remainder of the incubation.

The lung cell samples were placed onto chamber slides and incubated at 37° for 1 hr to allow cells to adhere. The sample was then washed with PBS and fixed with 4% paraformaldehyde. The samples were then stained for LAMP-1.

All animals were handled according to approved institutional animal care and use committee (IACUC) protocol #4946 at Washington State University.

### Transmission Electron Microscopy

BMDMs were inoculated with wildtype bacteria or Δ*dotU* ATc inducible *dotU* strain. 10 μg/ml of gentamicin was added to the BMDMs at 2 hr post inoculation. The cells were then incubated for a total of 20 hr. At 20 hr post inoculation, BMDMs seeded on a Nunc Thermomax plastic coverslip (ThermoFisher Scientific) were inverted onto the infected cells for synchronized transfer. The cells were incubated with recipient cells for 1 hr, purified and fixed for 1 hr with 2% paraformaldehyde and 2% glutaraldehyde in 0.1M cacodylate buffer (pH 7.2).

The BMDMs were microwave fixed using a Pelco Biowave Pro 36500 Laboratory Microwave System for 1 min at 350 Watts with a temperature restriction set to 38°C. The cells were rinsed 0.1M Cacodylate buffer 2X, 10 min each and 1X in distilled water for 10 min. The cells were incubated in 2% $OsO_4$ and 1.5% potassium ferrocyanide, 2 mM $CaCl_2$ in 0.05M cacodylate buffer (pH 7.2). They were rinsed 3X in distilled water, 10 min each and dehydrated in progressive graded ethanol series (10 min each step). The ethanol was replaced with acetone and the cells were infiltrated in 1:1 acetone and Spurr's resin overnight on a rotator. The acetone was evaporated over 6 hr. Fresh 100% resin was added and left to infiltrate overnight on a rotator, this step was repeated two more times. The cells were embedded and cured for 24 hr at 70°C. Sectioning, staining and imaging were done as described in *Froelich et al. (2011)*.

### Statistics

The statistical test for each experiment is listed in the figure legend. All details about the number of replicates, experiments, and cells analyzed are included in the figure legends.

The only cells excluded from our microscopy analyses were cells that had clearly undergone efforocytosis (large vacuole with DAPI stained nucleus in vacuole) and highly infected cells that were in a different focal plane on top of adherent cells because these are almost certainly donor cells that migrated to the coverslip.

## Acknowledgements

We would like to thank Daniel Mullendore at the Franceschi Microscopy and Imaging Core at Washington State University for preparing the samples for transmission electron microscopy and assistance in acquiring the images. We would like to thank Amanda Foreman for creating the graphical summary *Figure 7*.

## Additional information

### Funding

| Funder | Grant reference number | Author |
| --- | --- | --- |
| National Institute of Allergy and Infectious Diseases | AI082870 | Thomas H Kawula |

The funders had no role in study design, data collection and interpretation, or the decision to submit the work for publication.

### Author contributions
Shaun P Steele, Conceptualization, Investigation, Methodology, Writing—original draft, Writing—review and editing; Zach Chamberlain, Jason Park, Validation, Investigation; Thomas H Kawula, Conceptualization, Supervision, Funding acquisition, Writing—review and editing

### Author ORCIDs
Shaun P Steele (iD) http://orcid.org/0000-0002-3760-329X
Jason Park (iD) http://orcid.org/0000-0002-9050-5277
Thomas H Kawula (iD) http://orcid.org/0000-0001-7526-5159

### Ethics
Animal experimentation: All animals were handled according to approved institutional animal care and use committee (IACUC) protocol #4946 at Washington State University.

### Decision letter and Author response
Decision letter https://doi.org/10.7554/eLife.45252.021
Author response https://doi.org/10.7554/eLife.45252.022

## Additional files

### Supplementary files
• Transparent reporting form
DOI: https://doi.org/10.7554/eLife.45252.018

### Data availability
All data generated and analyzed in this study are included in the manuscript.

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
