## [Decision Letter]

Thank you for submitting your article Francisella tularensis enters a double membraned endosome following cell-cell transfer" for consideration by *eLife*. Your article has been reviewed by three peer reviewers, and the evaluation has been overseen by a Reviewing Editor and Wendy Garrett as the Senior Editor. The following individual involved in review of your submission has agreed to reveal their identity: Alain Charbit.

The reviewers have discussed the reviews with one another and the Reviewing Editor has drafted this decision to help you prepare a revised submission.

In the present study, you characterize in greater details the mechanism of trogocytosis that you reported previously where Francisella disseminates from macrophage to macrophage through exchange of cytosolic and membrane contents of infected cells with their non-infected neighbours. In this follow-up article, you describe that the recipient macrophages phagocytose parts of infected macrophages, through a process that you name "merocytophagy", leading to the formation of a double membrane phagosome in recipient cells. Maturation of the Francisella containing phagosome is very similar after merocytophagy or phagocytosis of extracellular bacteria, and F. tularensis escapes from this membrane-bound compartment to reach the cytosol using its Type 6 Secretion System.

The reviewers all agreed that the findings are interesting and important for the researchers working on intracellular bacteria. However they request to see a few more experimental additions to your work and stylistic changes before it can be accepted for publication.

Essential revisions:

1) The in vivo part of the work needs to be strengthened to prove the relevance of the phenomenon. Please provide additional images to Figure 6B and a quantification of such events vs. phagocytosis of extracellular Francisella.

2) Provide transmission electron micrograph images of wild-type (in addition to the already reported ones for inducible T6SS bacteria in Figure 5C-D) bacteria inside double membrane structures in purified recipient cells. Whereas the reviewers appreciated the elegant trick provided by the inducible T6SS, it would be worrisome if such events could not be captured for wild-type bacteria.

3) Improve clarity of the manuscript with an emphasis on how the experiments were carried out as opposed to focus solely on conclusions. For example, how were the recipient cells distinguished by EM? The paragraphs 'Multi-bacterial FCVs require Cell-Cell Contact" and "Transferred bacteria are enclosed in a double membrane structure containing membrane from the donor cells" are confusing and should be rewritten. Moreover, it would be helpful if the authors could provide information in the text as to what allows them to identify the object being engulfed as a bacterium. Also, the authors should comment on the fact that the engulfed object appears fragmented in some sections.

---

## [Author Response]

Essential revisions:1) The in vivo part of the work needs to be strengthened to prove the relevance of the phenomenon. Please provide additional images to Figure 6B and a quantification of such events vs. phagocytosis of extracellular Francisella.

For the in vivostudy, we do not yet have the tools to definitively differentiate between phagocytosis of bacteria and merocytophagy in vivo. Instead, we estimated the minimum rate of merocytophagy to be about 20% based on the transfer of large clusters of bacteria that remain in the same vacuole. This estimate does not include merocytophagy of individual bacteria or cells where the LAMP-1 staining was not definitive. We anticipate that more accurate estimates will emerge as tools to study merocytophagy improve. This addition is in the subsection “FCVs containing multiple bacteria occur in a mouse infection model”.

2) Provide transmission electron micrograph images of wild-type (in addition to the already reported ones for inducible T6SS bacteria in Figure 5C-D) bacteria inside double membrane structures in purified recipient cells. Whereas the reviewers appreciated the elegant trick provided by the inducible T6SS, it would be worrisome if such events could not be captured for wild-type bacteria.

We added transmission electron micrographs of wildtype bacteria inside of spacious, double-membraned vacuoles. We added images of the wildtype bacteria in vacuoles 40 minutes after transfer to both Figure 5 and Figure 5—figure supplement 5. The transferred cytosolic content of the vacuoles appears slightly different between the different strains. This is likely due to wildtype modifying the phagosome which is discussed in the last paragraph of the subsection “Transferred bacteria are enclosed in a multi-membranous structure”.

3) Improve clarity of the manuscript with an emphasis on how the experiments were carried out as opposed to focus solely on conclusions. For example, how were the recipient cells distinguished by EM? The paragraphs 'Multi-bacterial FCVs require Cell-Cell Contact" and "Transferred bacteria are enclosed in a double membrane structure containing membrane from the donor cells" are confusing and should be rewritten. Moreover, it would be helpful if the authors could provide information in the text as to what allows them to identify the object being engulfed as a bacterium. Also, the authors should comment on the fact that the engulfed object appears fragmented in some sections.

We made the requested editorial changes. We added a diagram (Figure 1F) to graphically explain the transfer assay and recipient cell purification used for the vast majority of our experiments. The two unclear paragraphs were rewritten for clarity (subsection “Transferred bacteria are enclosed in a multi-membranous structure” and subsection “Cell-Cell Contact is required for FCVs containing multiple bacteria to form”) and our interpretation of the TEM images showing bacterial transfer in the act has been added to the last paragraph of the subsection “Macrophages phagocytose portions of infected cells to acquire both bacteria and cytosolic material”. Our guideline for identifying bacteria are in the third paragraph of the subsection “Transferred bacteria are enclosed in a multi-membranous structure”. The title was changed as requested. The section describing how the type VI secretion system results fit into the field was re-written in the second paragraph of the Discussion. The specific coloring and orientation of images was also changed to fit the reviewers’ requests (Figure 1A and Figure 5B).